# Ubiquitination Is a Novel Post-Translational Modification of VMP1 in Autophagy of Human Tumor Cells

**DOI:** 10.3390/ijms241612981

**Published:** 2023-08-19

**Authors:** Felipe J. Renna, Juliana H. Enriqué Steinberg, Claudio D. Gonzalez, Maria Manifava, Mariana S. Tadic, Tamara Orquera, Carolina V. Vecino, Alejandro Ropolo, Daniele Guardavaccaro, Mario Rossi, Nicholas T. Ktistakis, Maria I. Vaccaro

**Affiliations:** 1Instituto de Bioquimica y Medicina Molecular Prof Alberto Boveris (IBIMOL), CONICET, Universidad de Buenos Aires, Buenos Aires C1113AAC, Argentina; fjrenna@ffyb.uba.ar (F.J.R.);; 2Instituto de Investigaciones en Medicina Traslacional (IIMT), CONICET, Universidad Austral, Pilar C1006ACC, Argentina; 3Department of Biotechnology, University of Verona, 37134 Verona, Italy; 4Signalling Programme, Babraham Institute, Cambridge CB22 3AT, UK

**Keywords:** autophagosome, tumor cells, DFCP1, VMP1

## Abstract

Autophagy is a tightly regulated catabolic process involved in the degradation and recycling of proteins and organelles. Ubiquitination plays an important role in the regulation of autophagy. Vacuole Membrane Protein 1 (VMP1) is an essential autophagy protein. The expression of VMP1 in pancreatic cancer stem cells carrying the activated Kirsten rat sarcoma viral oncogene homolog (KRAS) triggers autophagy and enables therapy resistance. Using biochemical and cellular approaches, we identified ubiquitination as a post-translational modification of VMP1 from the initial steps in autophagosome biogenesis. VMP1 remains ubiquitinated as part of the autophagosome membrane throughout autophagic flux until autolysosome formation. However, VMP1 is not degraded by autophagy, nor by the ubiquitin–proteasomal system. Mass spectrometry and immunoprecipitation showed that the cell division cycle protein cdt2 (Cdt2), the substrate recognition subunit of the E3 ligase complex associated with cancer, cullin–RING ubiquitin ligase complex 4 (CRL4), is a novel interactor of VMP1 and is involved in VMP1 ubiquitination. VMP1 ubiquitination decreases under the CRL inhibitor MLN4924 and increases with Cdt2 overexpression. Moreover, VMP1 recruitment and autophagosome formation is significantly affected by CRL inhibition. Our results indicate that ubiquitination is a novel post-translational modification of VMP1 during autophagy in human tumor cells. VMP1 ubiquitination may be of clinical relevance in tumor-cell-therapy resistance.

## 1. Introduction

Macroautophagy (commonly called autophagy) is a physiological catabolic process that allows protein degradation and organelle turnover through lysosomal degradation. During this process, the autophagosome, a specific double-membrane vesicle, is formed. It sequesters the cargo molecules driving them to the lysosome for degradation by hydrolytic enzymes [1,2]. Since autophagy is essential for the maintenance of cellular homeostasis, diverse stimuli are integrated in a complex network of signals, leading to autophagy induction. In this way, the cell is able to respond to stressors such as amino acid starvation, low energy levels, reticulum stress, hypoxia, and oxidative stress by triggering autophagy to repair damaged structures or to obtain energy sources [2]. Mechanistically, the autophagy pathway is orchestrated by a large set of evolutionarily conserved proteins known as autophagy-related proteins (ATG), which allows autophagosome formation and the complete development of the process [3]. Given the high complexity of the molecular mechanisms that govern autophagy and the essentiality of this process, it presents a tight regulation. This regulation is not only related to the vast induction pathway network, but also to the activity of each singular ATG protein. Post-translational modifications play an important role in modulating protein interactions, protein destinations, and catalytic activity in enzymatic proteins [4]. Ubiquitination is one of the most complex and versatile post-translational modifications [5].

During autophagy induction, several stress signals are integrated by two protein kinases: mammalian target of rapamycin (mTOR) and AMP-activated protein kinase (AMPK). These signals lead to AMPK activation and/or mTOR inhibition [2]. The next step in the autophagic pathway is the activation of the ULK1 complex. This complex is integrated by the proteins ATG13, FAK-family-interacting protein of 200 kDa (FIP200), ATG101, and the protein kinase ULK1, and it works as a serine/threonine protein kinase leading to the phosphorylation and activation of downstream substrates [6]. Downstream from the ULK1 complex, phosphatidylinositol 3-kinase type 3 complex 1 (PI3KC3-C1) is composed of the proteins Beclin 1 (BECN1), ATG14L, serine/threonine protein kinase VPS15 (Vps15), and phosphatidylinositol 3-kinase VPS34 (Vps34), the latter of which is the catalytic subunit, which produces phosphatidylinositol 3-phosphate (PI3P) in the site where the autophagosome begins to sprout [7]. This event results in the recruitment of PI3P-binding proteins, such as double FYVE-containing protein 1 (DFCP1) and WD40-repeat phosphoinositide-interacting protein 2 (WIPI2), and it is followed by the recruitment of the ATG12–ATG5–ATG16L complex, which allows the lipidation of microtubule-associated protein 1 light-chain 3 (LC3) with phosphatidylethanolamine (PE) [1,2]. After LC3 lipidation, cargo molecules are enveloped, the membrane is elongated, and the autophagosome finally sprouts. In the last part of this process, the autophagosome is fused with the lysosome in a process that is dependent on syntaxin-17 (STX17) and the Ras-related protein Rab-7a (RAB7), forming the autolysosome and leading to cargo degradation [8,9].

Ubiquitination (or ubiquitylation) consists of a covalent modification with one or more molecules of ubiquitin. Ubiquitin is a 76-amino-acid-length protein that is highly conserved in eukaryotic organisms. Ubiquitination takes place through the sequential action of three enzymes: the ubiquitin-activating enzyme (E1), the ubiquitin-conjugating enzyme (E2), and the ubiquitin protein ligase (E3) [10,11]. Ubiquitination presents a huge versatility given that ubiquitin can be added as a simple molecule (monoubiquitination) or as polymeric ubiquitin chains, presenting diverse variants. Ubiquitination plays a crucial role in the regulation of autophagy. It exerts significant control over a wide array of autophagy-related proteins, either facilitating their proteasomal or autophagosomal degradation or activating their specific functions. [5]. Ubiquitination mechanisms are implicated in the regulation of every stage of the autophagic process, allowing a more precise and faster adaptation to the microenvironmental conditions. Moreover, ubiquitination is also implicated in cargo labelling for selective types of autophagy [5].

Vacuole Membrane Protein 1 (VMP1) is a 406-amino-acid transmembrane protein whose expression per se induces autophagy in mammalian cells [12,13]. VMP1 interacts with BECN1, allowing the recruitment of PI3KC3 complex 1 to the site of autophagosome formation [12,13] in the endoplasmic reticulum (ER)–plasma membrane (PM) contact sites [14]. VMP1 is essential for autophagy [12], given that in the absence of VMP1 expression, autophagosomes remain attached to the endoplasmic reticulum and are not able to continue with the autophagic flux [15]. Moreover, VMP1 has been found to be part of the membrane of isolated autophagosomes in the context of the cellular response to acute pancreatitis [16]. Recently, based on the results of in vitro reconstitution assays, VMP1 has been proposed to have lipid scramblase activity [17], which adds complexity to its function and increases its relevance even more as a vesicle-trafficking protein.

The role of VMP1 is remarkably important in the cellular response to diseases. VMP1 expression is not detectable in a healthy human pancreas, but it is markedly triggered in response to pancreatitis [12], and pancreatic cancer [18,19]. Under acute pancreatitis-induced stress, VMP1 expression triggers zymophagy, the selective autophagy of activated zymogen granules [16], and mitophagy, the selective autophagy of damaged mitochondria [20], thus playing a protective role and avoiding the progression of pancreatitis to a severe disease. In pancreatic cancer cells, VMP1 expression is induced by mutated KRAS [21]. We recently showed that VMP1 expression is activated by gemcitabine treatment through the E2F1 transcription factor, triggering autophagy and mediating chemotherapy resistance [18,19]. In addition, VMP1 is involved in the resistance to photodynamic therapy in colon cancer cells, in which its expression is triggered by the transcription factor hypoxia-inducible factor 1 alpha (HIF-1α) [22]. Therefore, VMP1 plays a relevant role in tumor pathophysiology. Given the important role of VMP1 in the autophagy pathway, in pancreatic pathophysiology, and in tumor-resistance mechanisms, VMP1 should be tightly regulated not only at the transcriptional level, but also at the post-translational one.

In this study, we used tumor cells as an experimental model to study the ubiquitination of VMP1. We identified the ubiquitination as a post-translational modification for VMP1, which does not label it for degradation in the proteasome or by autophagy. VMP1 is ubiquitinated early in autophagosome biogenesis and remains ubiquitinated as part of the autophagosome membrane throughout the autophagic flux until autolysosome formation. Moreover, Cdt2 (cell division cycle protein cdt2), the substrate recognition subunit of the cullin–RING ubiquitin ligase complex 4 (CRL4), interacts with VMP1. We found that the CRL4/Cdt2 E3 ligase is involved in VMP1 ubiquitination as a novel post-translational modification, which might be involved in the regulation of the autophagic pathway in human tumor cells.

## 2. Results

### 2.1. VMP1 Is Post-Translationally Modified by Ubiquitination

We performed an immunofluorescence experiment on ubiquitin in HeLa cells transfected with VMP1–GFP. The number of ubiquitin dots per cell in the VMP1–GFP-transfected cells was significantly higher in comparison to non-transfected or enhanced green fluorescent protein (GFP)-transfected cells, indicating that the expression of VMP1 induces ubiquitin recruitment (Figure 1a). These data were confirmed in other tumor cell lines, PANC-1 (Appendix A) and MCF-7 (Appendix A). In these images, we observed that the distribution of ubiquitin dots coincided with the VMP1 distribution. Then, we performed immunofluorescence experiments on ubiquitin in VMP1–GFP-transfected HeLa cells and determined the colocalization between VMP1 and ubiquitin. Figure 1b shows that the VMP1–GFP expression was significantly colocalized with ubiquitin (Ub). To evaluate whether VMP1 was ubiquitinated, we performed an immunoprecipitation experiment with anti-FLAG in lysates from cells co-transfected with VMP1–GFP and FLAG–Ub. The immunoblotting of eluates with anti–FLAG showed the presence of a large set of ubiquitinated proteins. Immunoblotting with anti-VMP1 showed several bands corresponding to VMP1–GFP and ubiquitinated VMP1–GFP, suggesting that VMP1 is polyubiquitinated. In addition, a band around 85 kDa that could be compatible with monoubiquitination appears in Figure 1c. Then, we performed an immunoprecipitation experiment using anti-V5 in lysates from cells co-transfected with VMP1–V5 and FLAG–Ub. Immunoblotting with anti-V5 showed bands corresponding to VMP1–V5 and ubiquitinated VMP1–V5. Immunoblotting with anti-FLAG showed bands corresponding to ubiquitinated VMP1–V5 (Appendix A).

It has been well established that the overexpression of ubiquitin can lead to non-physiological modifications of cellular proteins [23]. Therefore, we investigated whether VMP1 undergoes ubiquitination even under normal cellular conditions, without the overexpression of ubiquitin. To address this question, we conducted immunoprecipitation experiments using anti-GFP on lysates from cells transfected with VMP1–GFP. Subsequent immunoblotting with anti-Ub revealed distinct bands corresponding to ubiquitinated VMP1–GFP (Figure 1d). Additionally, we carried out immunoprecipitation using anti-FLAG on lysates from cells transfected with VMP1–FLAG. Remarkably, these experiments also revealed several bands corresponding to ubiquitinated VMP1–FLAG (Appendix A). These results demonstrate that VMP1 is modified by ubiquitination.

To investigate whether VMP1 was also modified by monoubiquitination, we concomitantly transfected cells with VMP1–GFP and Myc–Ub (wild type) or Myc–Ub (K0). Lysine-free ubiquitin (K0 Ub) consists of a ubiquitin molecule whose lysine residues are replaced by arginine residues, making the formation of chains impossible (Figure 2a). The cell lysates were immunoprecipitated with anti-Myc-linked protein G–Sepharose and the eluates were immunoblotted with anti-GFP. We found that VMP1 was only ubiquitinated in the presence of Myc–Ub (wild type), but not Myc–Ub (K0) (Figure 2b). These results indicate that VMP1 is not monoubiquitinated and confirm that VMP1 is polyubiquitinated.

### 2.2. Ubiquitinated VMP1 Is Not Degraded by Autophagy or Ubiquitin–Proteasome System

Given that the ubiquitination of VMP1 might regulate its degradation, we aimed to analyze the degradation pathways for VMP1. In the first place, we tested whether VMP1 is degraded by the proteasome. For this purpose, HEK293T cells were transfected with VMP1–GFP and treated with the proteasome inhibitor MG132 for 6 h. We observed that VMP1–GFP was not accumulated under proteasome inhibition (Figure 3a,b). As a control for proteasome blocking, we quantified the levels of total ubiquitinated proteins, which showed a significant increase under MG132 treatment (Figure 3a,c). To confirm the obtained result, we performed a cycloheximide (CHX) chase assay in VMP1–GFP-expressing cells treated with or without MG132. The results showed that the VMP1–GFP levels decreased over a similar time course in the presence or absence of MG132 (Figure 3d–g). As a set, these results indicate that ubiquitinated VMP1 is not degraded by the ubiquitin–proteasome system.

In the next step, we analyzed whether VMP1 is degraded by autophagy. HEK293T cells were transfected with VMP1–GFP and treated with chloroquine (CQ) over 24 h. VMP1–GFP was not accumulated in chloroquine-treated cells compared to non-treated cells (Figure 3h,i). Indeed, a small, but significant, decrease was found in VMP1–GFP. As a control of lysosome blocking, we showed that the autophagic protein LC3-II levels (marker of autophagic degradation) significantly increased under the chloroquine treatment (Figure 3h,j). The absence of VMP1 accumulation in CQ-treated cells suggests that the ubiquitination of VMP1 does not drive it to degradation by autophagy. Then, we performed a cycloheximide chase assay in CQ-treated cells. We observed that the time course decrease in VMP1 levels under the CHX treatment (Figure 3k,l) was similar to the results obtained in cells without the CQ treatment (Figure 3d,e). These results indicate that ubiquitinated VMP1 is not degraded by autophagy.

### 2.3. Ubiquitinated VMP1 Is Involved in the Autophagic Process

So far, we know that VMP1 is modified by ubiquitination and that ubiquitination does not initially lead to VMP1 degradation by proteosomes or by autophagy. A large set of data shows that ubiquitination is crucial for the regulation of autophagy-related protein functions [5]. We therefore hypothesized that VMP1 ubiquitination could be related to its involvement in autophagy. We performed an immunofluorescence experiment on LC3 and ubiquitin in VMP1–GFP-transfected cells and we found that the LC3 dots were mostly associated with ubiquitin and VMP1. Interestingly, in a perinuclear zone where VMP1 was not associated with ubiquitin, it did not recruit LC3 (Figure 4a,b). Figure 4c shows the quantification of the LC3 dots associated with VMP1. While 76% of the LC3 dots were associated with VMP1 and ubiquitin (VMP1–Ub–LC3), only two percent of them were associated with non-ubiquitinated VMP1 (VMP1–LC3). It is noteworthy that almost all the VMP1 dots that colocalized with LC3 were associated with ubiquitin. To further validate the association between VMP1, Ub, and LC3, we conducted immunoprecipitation experiments. We co-transfected cells with LC3 fused to red fluorescent protein (RFP–LC3), VMP1–GFP, and FLAG–Ub, and subsequently, the lysates were subjected to immunoprecipitation using anti-GFP magnetic beads. Notably, the immunoblotting of the eluates with anti-LC3 demonstrated the presence of RFP–LC3-II (Figure 4d). Moreover, when probed with anti-FLAG, several bands corresponding to ubiquitinated VMP1-GFP were clearly observed (Figure 4d). This outcome significantly supports the association of ubiquitinated VMP1 with LC3-II, the lipidated form of LC3 that is known to be associated with autophagic structures.

We then used GFP–DFCP1 as a marker of omegasomes in VMP1–DsRed-transfected cells that were immunolabeled with anti-Ub. We found that, while 35% of VMP1 associated with ubiquitin co-distributed with double FYVE-containing protein 1 (DFCP1), DFCP1 co-distributed with VMP1 only when VMP1 was associated with ubiquitin (VMP1–Ub–DFCP1) (Figure 4e,f). These findings suggest that VMP1 is ubiquitinated from the initial steps of autophagosome biogenesis.

We next studied whether VMP1 remains ubiquitinated in the advanced steps of autophagy. By immunolabeling LAMP2 (a lysosome marker) in VMP1–GFP-transfected cells, we found that VMP1 significantly colocalized with the lysosomes labeled with LAMP2 (Figure 5a,b). Moreover, in cells concomitantly transfected with VMP1–DsRed and LAMP1–YFP and immunolabeled with anti-LC3, we found that VMP1 co-distributed with LAMP1 and LC3 (Figure 5c,d). In addition, VMP1–DsRed- and GFP–DFCP1-transfected cells that were labeled with a lysotracker showed that most of the VMP1 dots associated with lysosomal structures were not concomitantly associated with the omegasome (Figure 5e,f). These results suggest that VMP1 is present in the autolysosome. Finally, VMP1–DsRed- and LAMP1–YFP-co-transfected cells were immunolabeled with anti-ubiquitin. Figure 5g,h show that the VMP1 present in the autolysosome was associated with ubiquitin, since 98% of the VMP1 dots that co-distributed with LAMP1 were associated with ubiquitin dots (VMP1–Ub–LAMP1). These results show that VMP1 remains ubiquitinated in the autophagosome membrane after the fusion between the autophagosome and the lysosome.

### 2.4. The E3 Ligase Complex CRL4/Cdt2 Is Involved in the Ubiquitination of VMP1

The cullin-RING E3 ubiquitin ligase CRL4/Cdt2 is known for being a main modulator of genome stability, targeting important cell cycle proteins for degradation, such as the CDK inhibitor p21 and the chromatin-licensing and DNA replication factor 1 (Cdt1) [24]. A schematic representation of this complex is shown in Figure 6a, where we can see the protein Cdt2 working as the substrate recognition subunit through directly binding to the protein to be ubiquitinated. Although the role of Cdt2 as a regulator of the cell cycle has been widely studied, little is known about its possible extranuclear roles. To find new CRL4/Cdt2 substrates, HEK 293T cells were transfected with FLAG–HA–Cdt2, where Cdt2 was purified by tandem affinity purification from soluble and chromatin-linked fractions. These fractions were separated and independently analyzed using the mass spectrometry approach. Conspicuously, VMP1 was found to be a new Cdt2 interactor in the soluble fraction, but not in the chromatin-linked one (Figure 6b). To confirm the Cdt2–VMP1 interaction, we performed an immunoprecipitation experiment with anti-FLAG in VMP1–FLAG- and Myc–Cdt2-transfected cells. We found Myc–Cdt2 in the eluates, indicating the VMP1–Cdt2 interaction (Figure 6c). Moreover, we found endogenous VMP1 in the eluates from the FLAG–Cdt2-transfected cells immunoprecipitated with anti-FLAG (Figure 6d), confirming the interaction between these two proteins.

Given the essential cell-cycle-regulatory function of Cdt2, it is not a surprise that it is mainly localized in the nucleus. Since the high colocalization found between VMP1 and Ub takes place mainly in the cell cytoplasm, we hypothesized that if Cdt2 participates in VMP1 ubiquitination, a fraction of Cdt2 should be translocated to the extranuclear region when VMP1–GFP is expressed. To test this hypothesis, we performed an immunofluorescence experiment with anti-FLAG in FLAG–Cdt2-transfected cells in the presence or absence of a VMP1–EGFP transfection. While cells without a VMP1–GFP transfection showed mainly a nuclear Cdt2 localization, cells with VMP1–GFP showed an extranuclear localization and co-distribution between VMP1 and Cdt2 in the cytoplasm (Appendix A). We then performed an immunofluorescence experiment to detect endogenous Cdt2 in cells with or without VMP1–GFP expression, and the translocation of Cdt2 under VMP1 expression was confirmed. While cells without a VMP1–GFP transfection showed mainly a nuclear localization of Cdt2 (Figure 6e), cells expressing VMP1–GFP showed mainly an extranuclear localization of Cdt2 (Figure 6f). This last observation was quantified by measuring the integrated intra- and extranuclear fluorescence signal. The ratio of the extranuclear/intranuclear signal was increased significantly in cells expressing VMP1–GFP (Figure 6g). Taken together, these results indicate that VMP1 interacts with Cdt2 and suggest that the ubiquitin ligase CRL4/Cdt2 might catalyze the ubiquitination of VMP1. To test whether the VMP1–Cdt2 interaction might be involved in the ubiquitination of VMP1, we overexpressed FLAG–Cdt2 in cells that co-expressed VMP1–GFP and Myc–Ub. The lysates of these cells were immunoprecipitated with anti-Myc linked to protein G–Sepharose. In the immunoblot with anti-GFP, we observed that the ubiquitinated VMP1 increased under FLAG–Cdt2 overexpression (Figure 6h). Finally, we performed an IP of Myc–Ub in VMP1–GFP-transfected cells treated with MLN4924. MLN4924 is a neddylation inhibitor that prevents the activation of CRL E3 ligases [25]. We found that VMP1 ubiquitination decreased in the presence of the CRL ligase inhibitor (Figure 6h). Taken together, these results indicate that VMP1 interacts with Cdt2 and suggest that the ubiquitin ligase CRL4/Cdt2 can catalyze the ubiquitination of VMP1.

### 2.5. VMP1 Recruitment and Autophagosome Formation Are Affected under Inhibition of CRL Ligases

Here, we present evidence suggesting that VMP1 ubiquitination is not linked to its degradation. Additionally, we identified the involvement of the protein Cdt2 in VMP1 ubiquitination. We conducted further experiments to investigate the effects of inhibiting CRL ligases on VMP1-mediated autophagosome formation. To inhibit CRL ligases, we used MLN4924, which has previously been reported to induce autophagy after more than 16 h of treatment [26,27,28]. Therefore, we tested if MLN4924, applied for 5 h, affects autophagy in basal conditions. We treated HeLa cells with MLN4924 and performed immunofluorescence of LC3 (Figure 7a). We quantified the LC3 dots and did not find significant differences compared to the control (Figure 7b). To further confirm that MLN4924 does not affect the levels of basal autophagy when applied for 5 h, we treated HEK293T cells with MLN4924 and performed a Western blot of LC3 (Figure 7c). We did not observe significant changes between MLN4924-treated and control cells (Figure 7d) in these conditions.

In another set of experiments, we performed immunofluorescence of LC3 in MLN4924-treated and control pLenti-VMP1-GFP-transfected cells (Figure 7e). Figure 7f shows a significant decrease in the number of VMP1 dots per cell in the MLN4924-treated cells compared to control cells. As shown in Figure 7g, there was a significant reduction in LC3 dots per cell induced by VMP1 expression under MLN4924 treatment. These results suggest that the inhibition of CRLs by MLN4924 affects VMP1-mediated autophagy. To further confirm that MLN4924 inhibits VMP1-mediated autophagy, we transfected HEK293T cells with pLenti-VMP1-GFP, treated them with MLN4924, and performed a Western blot of LC3 (Figure 7h). We found that LC3-II levels significantly decreased under MLN4924 treatment compared to controls (Figure 7i).

## 3. Discussion

Autophagy is an essential catabolic process that is fundamental for the maintenance of cellular homeostasis. Dysfunctional autophagy is directly related to pathologic occurrence. Diabetes, cardiovascular diseases, neurodegenerative disorders, and cancer have been linked to autophagy alterations as part of their pathophysiological mechanisms [29]. Regarding cancer, a large set of data showed a complex and contradictory role of autophagy in the progression of the disease. In early stages, autophagy played a protective role, in which it avoided the accumulation of damaged cellular components and enhanced immunosurveillance, contributing to tumor suppression. However, in advanced cancers, autophagy acted to promote tumor survival and growth, helping the tumor cope with adverse conditions such as hypoxia and nutritional deficiency [30]. Indeed, several studies have demonstrated the important role of autophagy in chemotherapy resistance [31]. Previously, we found that the expression of the transmembrane protein VMP1 triggers autophagy in human tumor cells [21]. VMP1-mediated autophagy is induced by gemcitabine through the transcription factor E2F1 in pancreatic cancer cells and by photodynamic therapy through HIF-1α in colon cancer cells. Both pathways of VMP1 transcriptional regulation promote tumor cell resistance to anticancer treatments [19,22]. In this work, we focus on the post-translational regulation of VMP1 by the ubiquitin system, which is highly implicated in the regulation of metabolic reprogramming in cancer cells [32]. In addition, ubiquitination plays an important role in modulating autophagy, and managing this regulatory mechanism could be of relevance for cancer therapy. We have identified ubiquitination as a novel post-translational modification of the autophagy-related transmembrane protein VMP1. Our results indicate that VMP1 is ubiquitinated in the early steps of autophagosome biogenesis and remains ubiquitinated as part of the autophagosome membrane throughout the autophagic flux until autolysosome formation. Moreover, we have shown that the E3 ligase substrate adaptor Cdt2, whose action is closely related to cell cycle regulation [33], is a new partner of VMP1 and is involved in VMP1 ubiquitination.

We have shown that VMP1 is modified by ubiquitination using immunofluorescence and immunoprecipitation techniques (Figure 1 and Appendix A). In the immunoprecipitation of FLAG–Ub (Figure 1c), we confirmed that VMP1 is ubiquitinated, since several bands above the molecular weight of VMP1–GFP were found in the immunoblot with anti-VMP1. Considering that the theoretical molecular weight for VMP1–GFP is 73.2 kDa and the weight for FLAG–Ub is 9,8 kDa, the band around 85 kDa that is also present in the immunoblot could indicate the monoubiquitination of VMP1, the theoretical MW of which would be 83 kDa. However, when we investigated monoubiquitination in Figure 2b, we did not find evidence for the ubiquitination of VMP1 with Myc–Ub (K0), a ubiquitin molecule that is unable to form ubiquitin chains. Therefore, we concluded that VMP1 is polyubiquitinated, but not monoubiquitinated. In this way, the 85 kDa band in Figure 1f might represent the first ubiquitin molecule in the polyubiquitination chain.

Immunofluorescence and immunoprecipitation assays, using diverse expression systems for VMP1 showed that VMP1 is ubiquitinated as soon as it is expressed (Figure 1). However, ubiquitination did not lead to the degradation of VMP1. Indeed, it was not accumulated after the proteasomes or lysosomes were blocked. Therefore, the early ubiquitination of VMP1 might not explain VMP1 degradation by the ubiquitin–proteasome system or by autophagy (Figure 3). Since ubiquitinated VMP1 is not broken down by autophagosomes or the proteasome system, it is important to note that certain de-ubiquitinases could be involved in the retrieval of non-ubiquitinated VMP1. In fact, we have previously found that the de-ubiquitinase probable ubiquitin carboxyl-terminal hydrolase FAF-X (USP9x) is an interactor of VMP1 in the context of the selective autophagy of zymogen granules during experimental acute pancreatitis [16]. On the other hand, under the inhibition of the autophagic flux (Figure 3h–j), a small but significant decrease in VMP1–GFP was found. This could be affected by translational regulation feedback in response to the inhibition of the autophagic flux. Given that VMP1 expression is triggered under acute stress conditions [12], a rapid regulation at the mRNA level is also feasible. In this way, under CHX treatment, we did not find changes between the CQ-treated and untreated cells. These findings show that a reduction in VMP1 levels after CQ inhibition only occurs when the translation is active, endorsing the hypothesis of a translational regulation feedback. According to our results, VMP1 is not cargo of autophagy. This observation is consistent with the results shown by Itakura and Mizushima [34], where only a few dots, formed by the cargo receptor p62, colocalized with the VMP1–GFP dots. Notably, after 16 h of CHX treatment, we found an important decrease in VMP1 levels, but this reduction was not avoided by CQ or MG132 treatments. Recent reports have demonstrated that some autophagy-related proteins and cargo molecules could plausibly be eliminated by secretory pathways [35,36]. Further studies would be necessary to fully elucidate VMP1 degradation mechanisms.

In this work, we demonstrated that VMP1 is ubiquitinated when it is involved in the autophagic process (Figure 4). Ubiquitinated VMP1 is associated with a significant LC3 recruitment (Figure 4a–c). Moreover, this association was validated by co-immunoprecipitation experiments in which VMP1 immunoprecipitated with ubiquitin and LC3-II (Figure 4d). The presence of specifically LC3-II (and not LC3-I) in the eluates demonstrates that ubiquitinated VMP1 is a part of the autophagic structures. Interestingly, we noted that, although most of the VMP1 is ubiquitinated (Figure 4c), there is a little portion that is not. This portion of non-ubiquitinated VMP1 is located principally in the perinuclear region (Figure 4a) and is poorly distributed with LC3 (Figure 4c). Therefore, it is proposed that only the ubiquitinated VMP1 would participate in autophagosome biogenesis. Indeed, in Figure 4e,f, we show that VMP1 is ubiquitinated when it co-distributes with the omegasome marker DFCP1.

We have previously demonstrated that VMP1 is a transmembrane protein in the autophagosome [16]. However, no data about the presence or absence of VMP1 in the autolysosome have been reported up to now. In this work, we found significant colocalization between VMP1 and the lysosome marker LAMP2 (Figure 5a,b). Moreover, a triple association between VMP1, LAMP1, and LC3 was found (Figure 5c,d), suggesting that VMP1 is present in the autophagosomal membrane throughout the autophagic flux. Indeed, when we investigated the proportion of VMP1 associated with lysosomes, we found that most of the VMP1 dots associated with lysosomes were not concomitantly associated with omegasomes (Figure 5e,f). These findings show for the first time that VMP1 is not only present in the autophagosome, but also remains a transmembrane protein of the autophagosome until its fusion with the lysosome. Importantly, we found a triple co-distribution between VMP1, LAMP1, and Ub (Figure 5g,h), indicating that VMP1 remains ubiquitinated when it arrives at the autolysosome.

Ubiquitination plays an essential role in autophagy regulation, since most of a long list of autophagy-related proteins are regulated by ubiquitination [5]. For instance, under poor nutrient conditions, Unc-51-like kinase 1 (ULK1) is modified by the E3 ligase TNF receptor (TNFR)-associated factor 6 (TRAF6), enhancing its stability and function [37]. In PI3KC3 complex 1, Vps34 and ATG14 ubiquitination leads to proteasome degradation [38,39]. BECN1 is K63-polyubiquitinated, allowing its autophagy recruitment [40], or K11- and K48-polyubiquitinated for proteasomal degradation [41,42]. WIPI2 ubiquitination leads to its proteasomal degradation, and it is mediated by the Cul4 family of ubiquitin ligases during mitosis [28]. Here, we analyzed VMP1 ubiquitination as a complete set; it is possible that more than one VMP1 ubiquitination site and different types of VMP1 ubiquitination occur, and these modifications might regulate VMP1 functions. It is important to note that in this work, we used overexpressed VMP1 to simplify the study of VMP1 ubiquitination. Further studies will be necessary to determine how this modification intervenes in a physiological and/or pathological context and how it could be modulated.

Furthermore, we found that Cdt2 is a new VMP1 interactor (Figure 6b–d). Cdt2 works as the substrate recognition factor in the E3 ligase complex CRL4/Cdt2 [30,41]. The interaction between Cdt2 and VMP1 points at this E3 ligase as an important candidate to catalyze the ubiquitination of VMP1. So far, it is known that the complex CRL4/Cdt2 plays a fundamental role in the regulation of genome stability. CRL4/Cdt2 labels important cellular substrates for proteasomal degradation during the S phase of the cell cycle. Some of the known substrates of CRL4/Cdt2 are Cdt1, the CDK inhibitor p21, and the histone methyltransferase Set8 (KMT5A). All of them are key regulators of DNA replication, and their degradation prevents re-replication [24,33]. In the mass spectrometry performed in this work, we separated two fractions, the soluble fraction and the chromatin-attached fraction. VMP1 was found to be the main interactor present in the soluble fraction, but it was not present in the chromatin-attached fraction. This probably implies a novel function for Cdt2, given that the main described interactors up to now have been nuclear proteins. We indeed demonstrated that the overexpression of VMP1 triggers Cdt2 relocation (Figure 6e–g and Appendix A), and an important part of Cdt2 seems to leave the nucleus to interact with VMP1 when VMP1 is overexpressed. We demonstrated that Cdt2 is involved in VMP1 ubiquitination, since Cdt2 overexpression increased the ubiquitination of VMP1. Moreover, although MLN4924 could exert indirect effects [25], the inhibition of CRL ligases induced a reduction in VMP1 ubiquitination, further supporting the idea that the CRL4/Cdt2 complex is involved in the ubiquitination of VMP1 (Figure 6h). Notably, the translocation of Cdt2 from the nucleus to the cytoplasm induced by VMP1 expression is a totally novel event and supports our findings that VMP1 is regulated by ubiquitination as soon as it is expressed.

It is very interesting to note that both VMP1 and Cdt2 play important roles in tumor resistance. While VMP1 is related to poor patient outcomes in pancreatic cancer [18] and breast cancer [43], Cdt2 is associated with more aggressive hepatocellular carcinoma [44], gastric cancer [45], and melanoma [46]. Moreover, in pancreatic cancer, the inhibition of the CRL E3 ligases with MLN4924 has been shown to increase the sensitivity to radiotherapy [47]. Therefore, taking into account our previous work reporting that VMP1-mediated autophagy is involved in chemotherapy resistance [18,19], it is possible that the VMP1 and CRL4 molecular pathways might be cross-talking during tumor progression.

We observed that MLN4924 inhibits VMP1 ubiquitination after 5 h of treatment (Figure 6h). MLN4924 has been reported as an autophagy inducer when applied for 16 h or longer [25,26,27]. However, under 5 h of treatment, we demonstrated that MLN4924 alone does not induce autophagy (Figure 7a–d). Nevertheless, the inhibition of ubiquitination by MLN4924 treatment reduces the recruitment of VMP1 and LC3, which are induced by VMP1 expression (Figure 7e–i), suggesting that VMP1 ubiquitination regulates autophagy.

## 4. Materials and Methods

### 4.1. Cell Culture and Transfections

The HEK293T, HeLa, PANC-1, and MCF-7 cell lines were obtained from the American Type Culture Collection (ATCC). These cell lines were cultured in Dulbecco’s modified Eagle medium (Biological Industries, HaEmek, Israel) supplemented with 10% fetal bovine serum (Natocor, Córdoba, Argentina), 200 mM L-alanyl-L-glutamine dipeptide (Life Technologies™, Carlsbad, CA, USA), 100 U/μL penicillin, and 100 μg/μL streptomycin (Biological Industries, HaEmek, Israel). The cell lines were maintained at 37 °C and 5% CO_2_ in a humidified atmosphere. Mycoplasma contamination was checked monthly using DAPI staining and every six months using PCR. Transfections were performed using the X-treme GENE 9 DNA transfection reagent (Roche Applied Science, Penzberg, Germany) according to the manufacturer’s instructions. Transfection efficiency for the plasmid pEGFP-N1–VMP1 was tested in HEK293T and HeLa cells, resulting in 95% efficiency for HEK293T and 60% for HeLa cells. In this study, HEK293T cells were used for immunoprecipitation and Western blotting experiments, whereas HeLa cells were used for immunofluorescence experiments.

### 4.2. Treatments

For amino acid starvation, cells were washed two times with starvation medium (140 mM NaCl, 1 mM CaCl_2_, 1 mM MgCl_2_, 5 mM glucose, 1% BSA, and 20 mM HEPES (pH: 7.4), and incubated with the same starvation medium for 1 h. Pp242 (Santa Cruz Biotechnology, Santa Cruz, CA, USA) was dissolved in DMSO in a 1 mM stock solution and applied over 1 h at a 1 μM final concentration. Chloroquine diphosphate salt (Sigma Aldrich, Darmstadt, Germany) was dissolved in PBS in a 10 mM stock and applied over 16 or 24 h at a 50 μM final concentration, or over 2 to 8 h at a 100 50 μM concentration. MG132 (PeptaNova, Sandhausen, Germany) was dissolved in DMSO in a 10 mM stock solution and applied over 6 h at a 10 μM final concentration. Cycloheximide (CHX) (Sigma-Aldrich) was applied over 2, 4, 6, 8, or 16 h at a 100 μg/mL concentration. MLN4924 (Selleckchem, Houston, TX, USA) was dissolved in DMSO in a 5 mM stock solution and was applied over 5 h at a 1 μM concentration.

### 4.3. Expression Vectors

The plasmid pEGFP-N1–VMP1 was obtained by inserting the human *VMP1* sequence (NCBI reference sequence: NM_030938.5) into pEGFP-N1 (CLONTECH cat. #6085-1). The plasmid pcDNA4-B–hVMP1–V5/His was obtained by inserting the human *VMP1* sequence (NM_030938.5) into pcDNATM4/V5-HisB (Invitrogen catalog number V861-20). pLenti–hVMP1–FLAG was obtained by inserting the oligonucleotide MCS-flag into pENTR1A (Addgene Plasmid #17398). The new vector was named pENTR1F–flag2. The human *VMP1* sequence (NM_030938.5) was cloned into pENTR1F–flag2 using the GatewayTM LR ClonaseTM II (cat. No. 11791-020). pLVX-DsRed-Monomer-N1–hVMP1 was obtained by inserting the human *VMP1* sequence (NCBI reference sequence: NM_030938.5) into pLVX-DsRed-Monomer-N1 (CLONTECH cat. 632152). pLenti–VMP1–GFP was obtained by cloning the *VMP1* sequence (NCBI reference sequence: NM_030938.5) into pENTR1A–GFP-N2 (FR1) (Addgene Plasmid #19364). The new vector was recombined with pLenti PGK puro (Addgene Plasmid #19068). The plasmid pLenti-PGK-GFP was kindly gifted by Ana María Cuervo (Albert Einstein College of Medicine, New York). LAMP1–YFP and GFP–DFCP1 were obtained from Addgene (Plasmid #1816, 38269). FLAG–Ub was kindly gifted from Simona Polo (University of Milan). The plasmids pWC7–His–Myc–ubiquitin wt and pWC7–His–Myc–ubiquitin Kless were kindly gifted from Daniele Guardavaccaro (University of Verona). The plasmid pRFP-LC3 was kindly provided by Dr. Maria I. Colombo (Universidad Nacional de Cuyo, Consejo Nacional de Investigaciones Científicas y Técnicas (CONICET), Argentina). *CDT2* cDNA was cloned in pcDNA3.1 with a FLAG and a HA tag obtaining the plasmid pcDNA3–2xFLAG–2xHA–Cdt2. The plasmid pcDNA3–Myc–Cdt2 was obtained by inserting the human *CDT2* cDNA sequence into the pcDNA3.1 vector.

### 4.4. Western Blot Analysis

After different treatments and transfections, cells were lysed in lysis buffer (50 mM Tris-HCl (pH: 7.4), 250 mM NaCl, 25 mM NaF, 2mM EDTA, 0.1% Triton X-100, and Thermo Scientific™ Pierce Protease Inhibitors, Waltham, MA, USA). For LC3, Western blot cells were lysed in 50 mM Tris-HCl (pH: 8.0), 150 mM NaCl, 1% Triton X-100, and Thermo Scientific™ Pierce Protease Inhibitors. The protein concentration was determined using the bicinchoninic acid (BCA) protein assay reagent (Pierce, Thermo Scientific™, Waltham, MA, USA). An equal amount of protein was analyzed using SDS-PAGE and transferred to polyvinylidene fluoride PVDF membranes (0.22 μm pore size, Millipore, Burlington, MA, USA). The membranes were blocked with Odyssey Blocking Buffer (LI-COR Biosciences, Lincoln, NE, USA) at room temperature for 1 h and incubated with the corresponding primary antibodies overnight at 4 °C. The primary antibodies used were anti-VMP1 (1:1000; rabbit mAb #12978, Cell Signaling Technology, Danvers, MA, USA), anti-actin (1:2000; rabbit polyAb A2066; Sigma–Aldrich), anti-ubiquitin (1:1000; mouse mAb UBCJ2; Enzo, Farmingdale, NY, USA), anti-DYKDDDDK tag (1:6000; mouse mAb #8146, Cell Signaling Technology), anti-α-tubulin (1:4000; mouse mAb T5168; Sigma-Aldrich), anti-Myc (1/1000; rabbit polyclonal ab9106; ABCAM, Cambridge, UK), and anti-GFP (1/3000, rabbit pAb TP401, Torrey Pines Biolab, Secaucus, NJ, USA). After incubation, the membrane was washed four times with PBS containing 0.1% Tween-20 (PBS-T), washed twice with PBS, and then incubated with the corresponding IRDye secondary antibody (1:15,000, IRDye^®^ 680LT goat anti-rabbit IgG or 1:10,000, IRDye^®^ 800CW goat anti-mouse IgG, LI-COR) in Odyssey Blocking Buffer (LI-COR) for 2 h at room temperature. Finally, the membrane was washed four times with PBST, washed twice with PBS, and scanned with Odyssey^®^ SA (LI-COR).

For the LC3 Western blotting, the membranes were blocked at room temperature for 1 h with 1% bovine serum albumin in Tris-buffered saline (TBS) containing 0.1% Tween-20 (TBS-T). After blocking, the membranes were incubated with the primary antibody anti-LC3B (1:500, rabbit mAb #3868, Cell Signaling Technology) over 24 h at 4 °C. After the incubation, the membranes were washed four times with TBS-T and twice with TBS. Finally, they were incubated with an anti-rabbit HRP-conjugated (1:2500, Amersham NA934, GE Healthcare, Chicago, IL, USA) secondary antibody in 1% bovine serum albumin in Tris-buffered saline (TBS) containing 0.1% Tween-20 (TBS-T). The membranes were washed four times with TBST, washed twice with TBS, and incubated with PIERCE ELC Plus Western blotting substrate (Cat# 32134, Thermo Scientific, Waltham, MA, USA), according to the manufacturer’s instructions, and scanned with cDigit Blot Scanner (LI-COR).

FIJI software (ImageJ 1.54f) was used to determine the density of protein bands. The relative densitometry, normalized to actin, was expressed as the mean ± SEM of three different experiments.

### 4.5. Immunoprecipitation Assays

For anti-FLAG immunoprecipitations, the supernatants of lysed cells were incubated in shake tubes over 2 h at 4 °C with Anti-FLAG^®^ M2 magnetic beads. The beads were washed with lysis buffer and eluates were obtained through an addition of Laemmli Buffer.

For anti-V5 immunoprecipitations, the supernatants of lysed cells were incubated in shake tubes over 1 h at 4 °C with Pierce™ Protein G magnetic beads (Invitrogen, Carlsbad, CA, USA) previously attached to V5 tag antibodies (R960-25). The beads were washed with lysis buffer and eluates were obtained through an addition of Laemmli Buffer.

For anti-GFP immunoprecipitations, the supernatants of lysed cells were incubated in shake tubes over 1.5 h at 4 °C with Pierce™ Protein G magnetic beads (Invitrogen) previously attached to the GFP tag antibody (A-11120). The beads were washed with lysis buffer and eluates were obtained through an addition of Laemmli Buffer.

For anti-Myc immunoprecipitations, the supernatants of lysed cells were incubated in shake tubes over 16 h at 4 °C with c-Myc antibodies (9E10 M5546, Merck, Darmstadt, Germany). Then, the lysates were incubated in shake tubes over 1 h at 4 °C with recombinant protein G–Sepharose (Invitrogen). The Sepharose was washed with RIPA lysis buffer and eluates were obtained through the addition of Laemmli Buffer.

### 4.6. Mass Spectrometry

HEK293T cells were transfected with Flag–HA-tagged Cdt2 or an empty vector (PCDNA). The CSK-soluble fraction and chromatin cell extracts were subjected to immunoprecipitation with an anti-FLAG antibody and eluted by competing with a FLAG peptide. Then, the material was immunoprecipitated with an anti-HA antibody. The mass spectrometry approach was used to identify new CRL4/Cdt2 substrates.

### 4.7. Immunofluorescence

For Hela, PANC-1, and MCF-7 cells, the following protocol was applied: The cells were grown on 12 mm round glass coverslips in 24-well plates. An amount of 3.5 × 10^4^ HeLa, 5 × 10^4^ MCF-7, or 2.5 × 10^4^ PANC-1 was seeded. The next day (or after two days, in the case of PANC-1), the cells were transfected. After 24 h, they were treated, fixed with 3.6% p-formaldehyde for 20 min, permeabilized for 5 min in Triton 0.1% in PBS, and blocked over 1 h. The glasses were incubated overnight with the primary antibody at 4 °C. The antibodies used were anti-ubiquitin (1:2000, mouse mAb UBCJ2; Enzo), anti-LC3 (1:1000, rabbit mAb #3868, Cell Signaling Technology), anti-FLAG (1:500; DYKDDDDK tag (9A3) mouse mAb; Cell Signaling Technology), and anti-Cdt2 (1:1000; rabbit polyclonal antibody NB100-40840; Novus Biologicals, Littleton, CO, USA). The next day, the glasses were incubated for 2 h at room temperature with secondary antibodies (1:800 Alexa Fluor 594 and 647 antibodies from Molecular Probes) and mounted in polyvinyl alcohol mounting medium with DABCO [48]. For lysotracker staining, the cells were incubated for 45 min with LysoTracker™ Blue DND-22 160 nM (Thermo Fisher Scientific, Waltham, MA, USA) and immediately observed and photographed using confocal microscopy in the absence of fixation.

For HEK 293T cells, 80% confluency cells were transfected into a 6-well plate. The next day, they were seeded on 16 mm round glass coverslips into 12-well plates. After 24 h, they were treated, fixed with 3.6% p-formaldehyde in PBS for 20 min, permeabilized with NETgel (150 mM NaCl, 5 mM EDTA, 50 mM Tris Cl (pH: 7.4), 0.05% (*v*/*v*) NP-40, 0.25% (*w*/*v*) gelatin (from bovine skin; Sigma, cat. No. G-6650), and 0.02% (*w*/*v*) sodium azide) containing 0.25% NP40, and stained for 30 min with the primary antibody. The antibodies used were anti-Ub (1:100; mouse monoclonal antibody FK2; ENZO), anti-LAMP2 (1:100; mouse monoclonal antibody; Developmental Studies Hybridoma Data Bank, Iowa City, IA, USA), and anti-ATG13 (1:100, mouse monoclonal antibody MABC46; Millipore, Burlington, MA, USA). After the primary antibody incubation, the cells were washed 3 times for 5 min with NETgel and stained for 30 min with goat anti-mouse IgG—Alexa Fluor 568. Mounting was performed with Aqua Poly Mount mounting medium (Polysciences, cat. No. 18606).

The sample observation and image acquisition were performed with a Zeiss Axio Imager D2 wide-field epi-fluorescence microscope equipped with a 63 1.4 NA lens, AxioCam HR CCD camera (Zeiss, Oberkochen, Germany), and HXP 120C metal halide light source (Zeiss), and an inverted LSM Olympus FV-1000 using an UPLSAPO 60X O NA: 1.35 objective.

### 4.8. Quantification of Colocalization

Pearson’s correlation coefficient for the selected ROI corresponding to the cell of interest was quantified using the plugin EzColocalization in the Fiji (Fiji Is Just ImageJ) open-source software (ImageJ 1.54f).

### 4.9. Quantification of Dots per Cell

The number of dots per cell was obtained through the 3D objects counter tool in the Fiji (Fiji Is Just ImageJ) open-source software (ImageJ 1.54f) [48].

### 4.10. Quantification of Dot Association

The images of HeLa transfected with VMP1–GFP and immunolabeled with anti-LC3 and anti-ubiquitin (with or without PP242 treatment) were analyzed using the Fiji (Fiji Is Just ImageJ) open-source software (ImageJ 1.54f). For the study of the association between LC3 or VMP1 and the other labelled proteins, we drew a 5 μm, 2-per-point grid over the composite image. For each LC3 or VMP1 point, the association with the signals in the other channels was judged using a high magnification.

### 4.11. Quantification of Ratio Extranuclear/Nuclear Localization

Using Fiji (Fiji Is Just ImageJ) (ImageJ 1.54f), we designed two ROIs per cell. The first one was located around the nucleus and the second one was around the complete cell but excluding the nucleus. The integrated density was measured for each ROI and the extranuclear/nuclear ratio was calculated.

### 4.12. Statistical Analysis

The data are expressed as the mean ± SEM. All the shown images are representative of three independent experiments. The statistical analyses of the data were performed using GraphPad Prism 8, as stated in the corresponding figure legends.

## 5. Conclusions

In the present study, we identified ubiquitination as a post-translational modification for VMP1. VMP1 ubiquitination does not lead to degradation by autophagy or the ubiquitin–proteasomal system. We demonstrated that VMP1 is ubiquitinated early in autophagosome biogenesis and remains ubiquitinated as part of the autophagosome membrane throughout autophagic flux. Moreover, VMP1 interacts with Cdt2, the substrate-recognition subunit of the CRL4/Cdt2 E3 ligase complex, which is involved in the ubiquitination of VMP1. Furthermore, we showed that VMP1 ubiquitination is part of the VMP1-mediated autophagy pathway schematically represented in Figure 8. We conclude that ubiquitination is a novel post-translational modification of VMP1 in the context of the VMP1-mediated autophagy of human tumor cells, and it might have clinical relevance in the resistance of cancer cells to therapy.

## Figures and Tables

**Figure 1 ijms-24-12981-f001:**
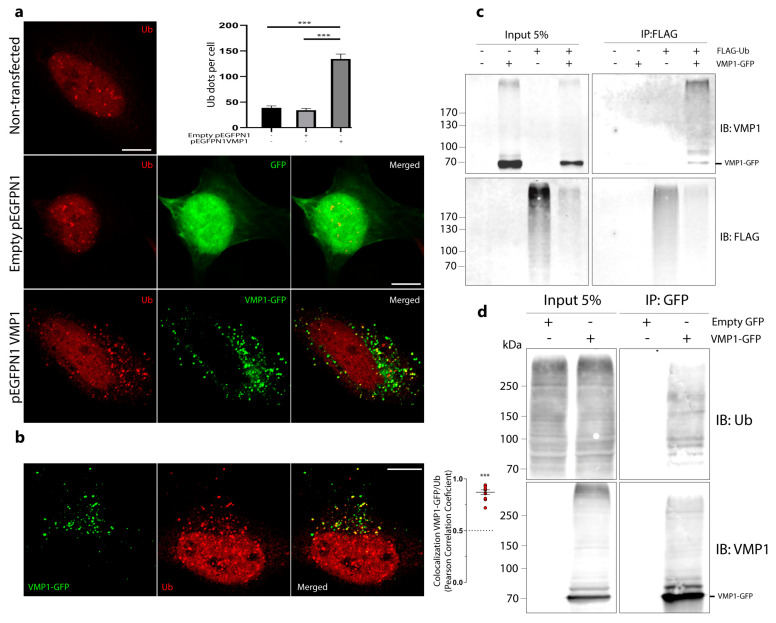
VMP1 is post–translationally modified by ubiquitination. (**a**) HeLa cells expressing empty pEGFPN1 or pEGFPN1 VMP1, or non–transfected HeLa cells were immunolabeled with anti–ubiquitin. Scale bars: 10 μm. The number of ubiquitin (Ub) dots per cell was quantified in at least 12 individual cells per condition in three independent experiments. The means per condition and the SEM are represented in the graphic. *** *p* < 0.001 according to a Kruskall–Wallis test. (**b**) HeLa cells were transfected with VMP1–GFP and immunolabeled with anti–ubiquitin. Scale bar: 10 μm. Colocalization between VMP1 and ubiquitin was quantified in 10 individual cells in three independent experiments. In the graphic, the red plots show Pearson’s correlation coefficient (PCC). *** *p* < 0.001 according to a two–tailed Student’s *t*–test vs. the theoretical mean, 0.5. (**c**) Lysates from HEK293T cells transfected with FLAG–Ub, VMP1–GFP, and the combination of both were immunoprecipitated with anti–FLAG magnetic beads and immunoblotted with anti–VMP1. Several bands over VMP1–GFP’s molecular weight (73 kDa) appeared in the eluate of co–transfected cells, indicating VMP1 ubiquitination. The image is representative of three independent experiments. (**d**) Lysates from HEK293T cells transfected with VMP1–GFP were immunoprecipitated with anti–GFP magnetic beads and immunoblotted with anti–Ub. Several bands over VMP1–GFP’s molecular weight (73 kDa) appeared in the eluate, indicating VMP1 ubiquitination.

**Figure 2 ijms-24-12981-f002:**
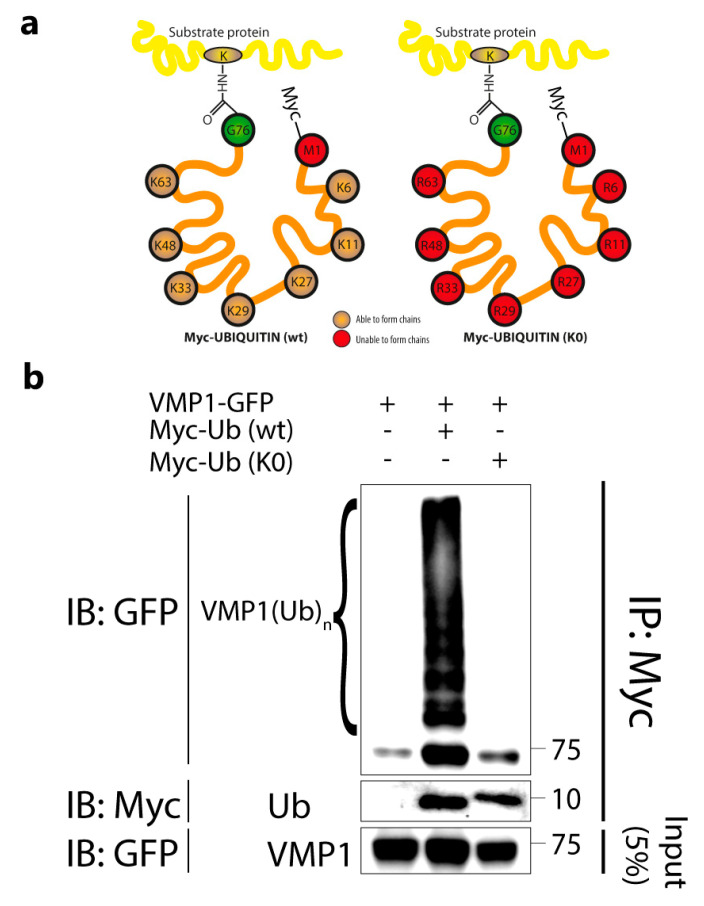
VMP1 is not monoubiquitinated with Myc–Ub (K0). (**a**) Schematic representation of Myc–Ub (K0) and Myc–Ub (wt) showing that Myc–Ub (K0) had its seven lysine residues replaced by arginine residues. In this way, Myc–Ub (K0) is only able to achieve monoubiquitination. (**b**) Lysates from HEK293T cells transfected with VMP1–GFP and Myc–Ub (wt) or Myc–Ub (K0) were immunoprecipitated with anti–Myc linked to protein G–Sepharose. The immunoblot with anti–GFP showed that there were no ubiquitination bands in the eluates from Myc–Ub (K0)–expressing cells, indicating that VMP1 is not monoubiquitinated. In contrast, eluates from Myc–Ub (wt)–expressing cells showed several bands corresponding to the polyubiquitination of VMP1. The image is representative of three independent experiments. K0 Ub: lysine–free ubiquitin; wt: wild type.

**Figure 3 ijms-24-12981-f003:**
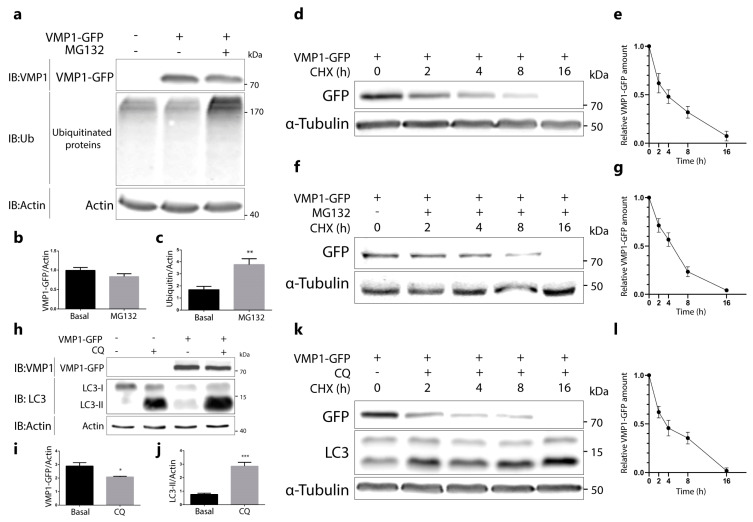
VMP1 is not degraded by the ubiquitin–proteasomal system, nor by autophagy. (**a**) Western blotting for VMP1, ubiquitin, and actin, where actin served as a loading control. HEK293T cells were transfected with VMP1–GFP and treated with or without the proteasome inhibitor MG132 (10 μM) over 6 h. The images are representative of three independent experiments. (**b**) Quantification of relative densitometry shows that VMP1–GFP (VMP1–GFP/actin) does not change significantly after treatment with MG132. (**c**) Relative densitometry (ubiquitin/actin) shows that ubiquitin levels increase significantly after MG132 treatment, indicating that the proteasome was effectively blocked. ** *p* < 0.01 vs. basal according to a Mann–Whitney nonparametric test; *n* = 3. (**d**) Western blotting for GFP and α–tubulin, where α–tubulin served as a loading control. HEK293T cells were transfected with VMP1–GFP and treated with 100 μg/mL of cycloheximide (CHX) over 0, 2, 4, 8, or 16 h. The images are representative of three independent experiments. (**e**) The graphic shows the means and SEMs of relative densitometry (VMP1–GFP/α–tubulin) in relation to the relative densitometry at 0 h; *n* = 3 for each experiment. (**f**) Western blotting for GFP and α–tubulin. HEK293T cells were transfected with VMP1–GFP and treated with CHX (100 μg/mL) and MG132 (10 μM) over 0, 2, 4, 8, or 16 h. The images are representative of three independent experiments. (**g**) The graphic shows the means and SEMs of relative densitometry (VMP1–GFP/α–tubulin) in relation to the relative densitometry at 0 h; *n* = 3 for each experiment. (**h**) Western blotting for VMP1, microtubule-associated protein 1 light–chain 3 (LC3), and actin, where actin served as a loading control. HEK 293T cells were transfected with VMP1–GFP and treated with or without 50 μM of the lysosome inhibitor chloroquine (CQ) over 24 h. The images are representative of three independent experiments. (**i**) Quantification of relative densitometry showed that VMP1–GFP (VMP1–GFP/actin) does not accumulate after treatment with CQ. In fact, VMP1–GFP significantly decreased in CQ–treated cells. * *p* < 0.05 vs. basal according to a Mann–Whitney nonparametric test; *n* = 3. (**j**) Relative densitometry (LC3–II/actin) showed that LC3–II levels significantly increased after CQ treatment, indicating that the lysosomal pathway was effectively blocked. *** *p* < 0.001 vs. basal according to a Mann–Whitney nonparametric test; *n* = 3. (**k**) Western blotting for GFP, LC3, and α–tubulin. HEK293T cells were transfected with VMP1–GFP and treated with CHX (100 μg/mL) and CQ (100 μM) over 0, 2, 4, or 8 h, or CHX 100 (μg/mL) and CQ (50 μM) over 16 h. The images are representative of three independent experiments. (**l**) The graphic shows the means and SEMs of relative densitometry (VMP1–GFP/α–tubulin) in relation to the relative densitometry at 0 h; *n* = 3 for each experiment.

**Figure 4 ijms-24-12981-f004:**
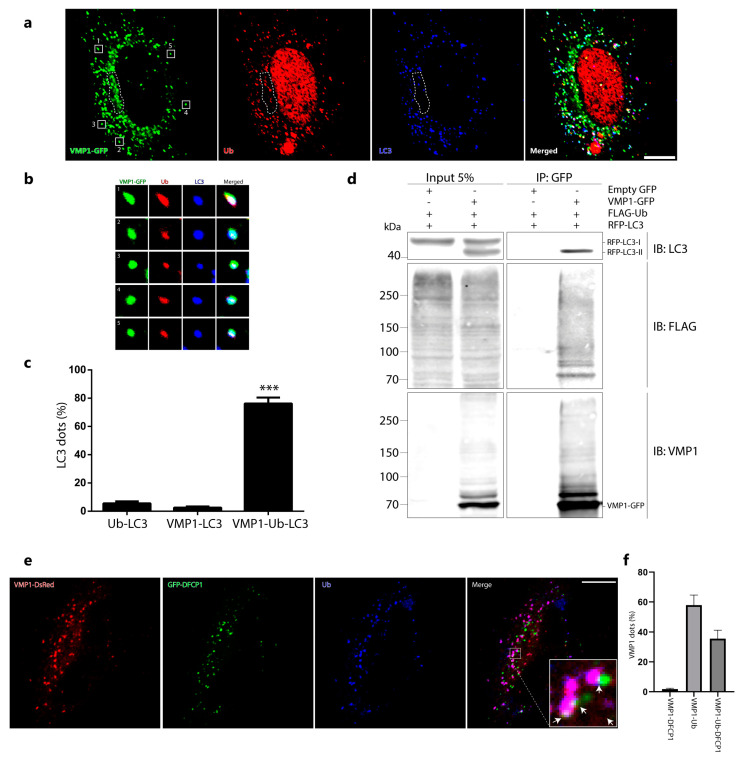
VMP1 associated with ubiquitin is involved in the autophagic process. (**a**) HeLa cells were transfected with VMP1–GFP and immunolabeled with anti-ubiquitin and anti–LC3. Scale bar: 10 μm. The images are representative of three independent experiments. The area bounded by the dotted line shows a perinuclear zone where the VMP1 that was not associated with Ub did not recruit LC3. (**b**) The square magnifications show examples of VMP1 dots that are associated with Ub and LC3. (**c**) Quantification of LC3 dots per cell associated with ubiquitin, VMP1, or ubiquitin and VMP1. *** *p* < 0.001 according to two–way ANOVA vs. Ub and VMP1 groups. *n* = 10 cells in three independent experiments. (**d**) Lysates from HEK293T cells that transfected with VMP1–GFP, FLAG–Ub and RFP–LC3 were immunoprecipitated with anti–GFP magnetic beads. The eluates were immunoblotted with anti–FLAG and anti–LC3. LC3 immunoblot shows a band of 41 kDa indicating the presence of RFP–LC3–II in the eluates. FLAG immunoblot shows the presence of ubiquitinated VMP1–GFP. (**e**) HeLa cells were transfected with VMP1–DsRed and GFP–DFCP1 and immunolabeled with anti–Ub. Scale bar: 10 μm. Arrows point to VMP1 dots that are associated with Ub and DFCP1. (**f**) Quantification of VMP1 dots per cell associated with DFCP1, ubiquitin, or ubiquitin and DFCP1. The graphic shows the means and SEMs. *n* = 10 cells in three independent experiments.

**Figure 5 ijms-24-12981-f005:**
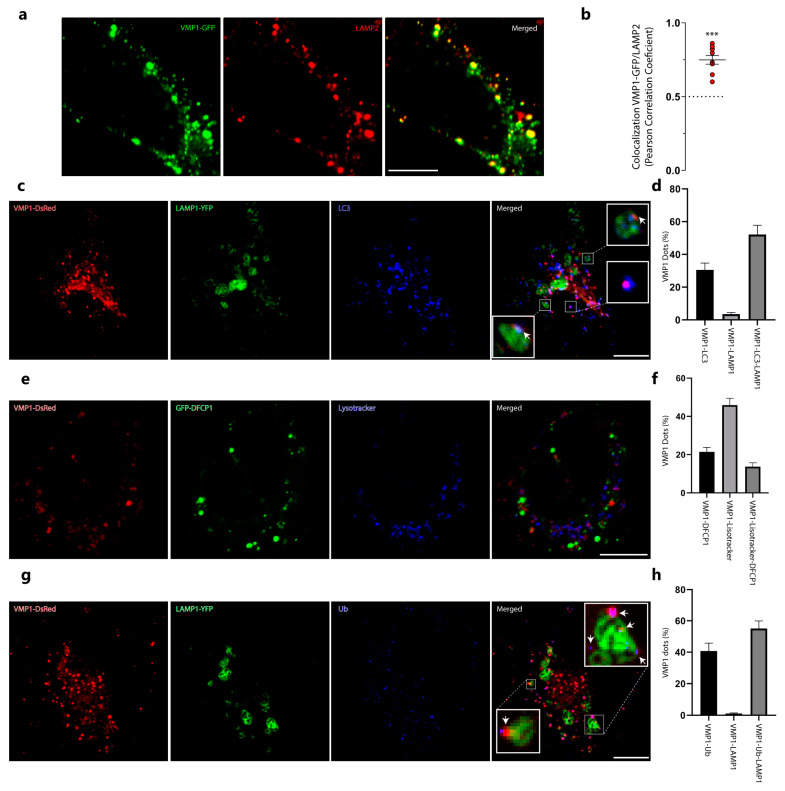
VMP1 remains ubiquitinated in the autophagic flux until it reaches the autolysosome. (**a**) HeLa cells were transfected with VMP1–GFP and immunolabeled with anti-LAMP2. Scale bars: 10 μm. (**b**) Colocalization between VMP1 and lysosome–associated membrane glycoprotein 2 (LAMP2) was quantified by Pearson’s correlation coefficient (PCC). Similar results were obtained in three independent experiments. The red plots show PCC for individual cells; *n* = 10. *** *p* < 0.001 according to a Kruskall–Wallis test. (**c**) HeLa cells were transfected with VMP1–DsRed and LAMP1–YFPs and immunolabeled with anti–LC3. Scale bar: 10 μm. Arrows point to VMP1 dots that are associated with lysosome–associated membrane glycoprotein 1 (LAMP1) and LC3. (**d**) Quantification of VMP1 dots per cell associated with LC3, LAMP1, and LC3–LAMP1. The graphic shows the means and SEMs. *n* = 10 cells in three independent experiments. (**e**) HeLa cells were transfected with VMP1–DsRed and GFP–DFCP1 and labeled with Lysotracker Blue DND–22. Scale bar: 10 μm. (**f**) The graphic shows the means and SEMs of the VMP1 dots associated with DFCP1, Lysotracker, or DFCP1 and Lysotracker. *n* = 10 cells in three independent experiments. (**g**) HeLa cells were transfected with VMP1–DsRed and LAMP1–YFPs and immunolabeled with anti–Ub. Scale bar: 10 μm. Arrows point to VMP1 dots that are associated with Ub and LAMP1. (**h**) Quantification of VMP1 dots per cell associated with ubiquitin, LAMP1, and Ub–LAMP1. The graphic shows the means and SEMs. *n* = 10 cells in three independent experiments. YFP: yellow fluorescent protein.

**Figure 6 ijms-24-12981-f006:**
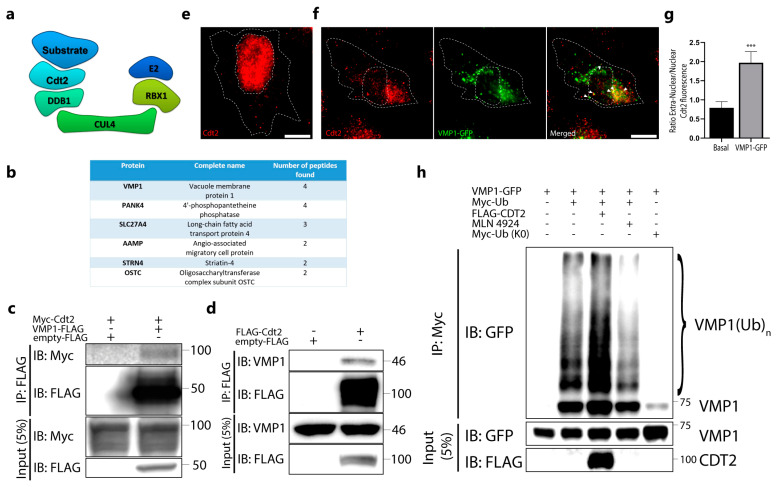
Cdt2 is involved in the ubiquitination of VMP1. (**a**) Schematic representation of the E3 ligase complex cullin–RING ubiquitin ligase complex 4 (CRL4)/Cdt2, in which the cell division cycle protein cdt2 (Cdt2) works as the substrate recognition subunit. (**b**) Table of main interactors of Cdt2 identified by MS of the sequential immunoprecipitation with anti–FLAG and anti–HA of soluble-fraction (non–chromatin–fraction) HEK293T cells transfected with FLAG–HA–Cdt2. VMP1 was found between the interactors. (**c**) Lysates from HEK293T cells transfected with Myc–Cdt2 and VMP1–FLAG were immunoprecipitated with anti–FLAG magnetic beads and immunoblotted with anti–Myc. (**d**) Lysates from HEK293T cells transfected with FLAG–Cdt2 were immunoprecipitated with anti–FLAG magnetic beads and immunoblotted with anti–VMP1. (**e**) HeLa cells were immunolabeled with anti–Cdt2. Scale bar: 10 μm. (**f**) HeLa cells were transfected with VMP1–GFP and immunolabeled with anti-Cdt2. Scale bar: 10 μm. Arrowheads point to VMP1 dots that co–distribute with Cdt2. (**g**) Endogenous Cdt2 localization was measured as the ratio between the extranuclear and intranuclear integrated density of fluorescence in VMP1–GFP– and non–transfected HeLa cells. *** *p* < 0.001 vs. control according to a Mann–Whitney nonparametric test. (**h**) Lysates from HEK293T cells transfected with VMP1–GFP and Myc–Ub with or without Cdt2–FLAG, and treated or not with MLN4924 (1 μM, 5 h), were immunoprecipitated with anti–Myc linked to protein G–Sepharose. The eluates were immunoblotted with anti–GFP. Ubiquitinated VMP1 increased in the presence of Cdt2–FLAG and decreased under treatment with MLN4924. Myc–Ub (K0) was used as a negative control. The images are representative of three independent experiments.

**Figure 7 ijms-24-12981-f007:**
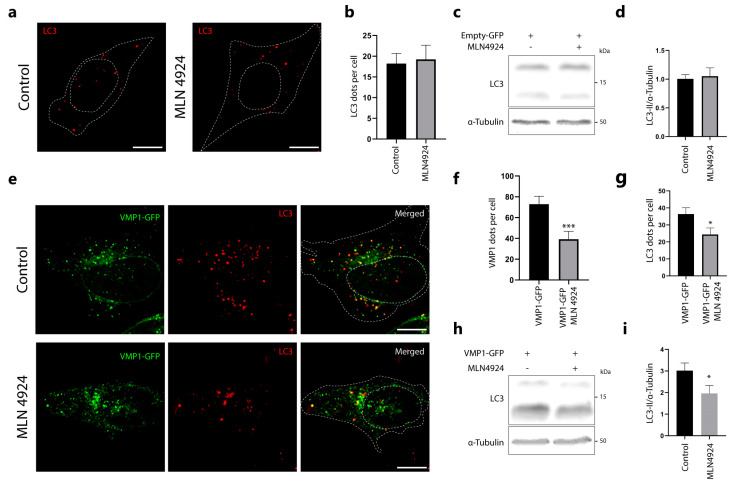
The distribution of VMP1 and its ability to trigger autophagy are both affected by treatment with MLN4924, a CRL inhibitor. (**a**) HeLa cells were treated with MLN4924 (1 μM, 5 h) or solvent. The cells were then immunolabeled with anti–LC3. Representative images are shown. Scale bar: 10 μm. (**b**) The number of LC3 dots per cell was quantified in both the control and MLN4924–treated cells. N = 13 cells per condition from three independent experiments. (**c**) Western blotting for LC3 and α–tubulin where α–tubulin served as a loading control. HEK293T cells were transfected with Empty–GFP and either treated with MLN4924 (1 μM, 5 h) or solvent. Images are representative of three independent experiments. (**d**) Quantification of relative densitometry shows LC3–II (LC3–II/α–tubulin) does not change significantly after treatment with MLN4924. N:3. (**e**) HeLa cells were transfected with pLenti–VMP1–GFP and either treated with MLN4924 (1 μM, 5 h) or solvent. The cells were then immunolabeled with anti–LC3. Representative images are shown. Scale bar: 10 μm. (**f**) The number of VMP1 dots per cell was significantly reduced in MLN4924–treated cells. The graph shows the means and SEMs of 18 control cells and 22 MLN4924–treated cells across three independent experiments. *** *p* < 0.001 vs. Control by Mann–Whitney test. (**g**) The number of LC3 dots per cell was also significantly reduced in MLN4924–treated cells compared to control. The means and SEMs are shown in the graph. The analysis was based on 18 control cells and 17 MLN4924–treated cells across three independent experiments. * *p* < 0.05 vs. Control by Mann–Whitney test. (**h**) Western blotting for LC3 and α–tubulin where α–tubulin served as a loading control. HEK293T cells were transfected with pLenti–VMP1–GFP and either treated with MLN4924 (1 μM, 5 h) or solvent. Images are representative of four independent experiments (**i**) Quantification of relative densitometry shows LC3–II (LC3–II/α–tubulin) significantly decrease after treatment with MLN4924. * *p* < 0.05 vs. Control by Student’s *t* test. N = 4.

**Figure 8 ijms-24-12981-f008:**
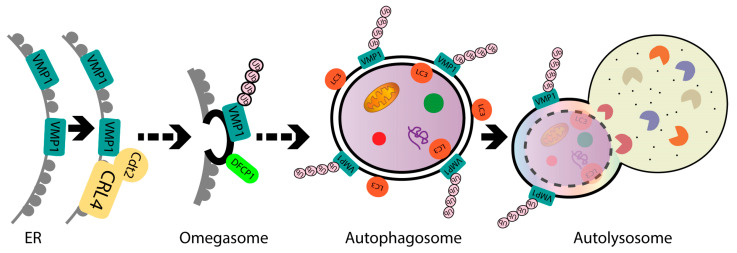
Schematic model. VMP1 is ubiquitinated in the initial steps of autophagosome biogenesis and remains ubiquitinated throughout the autophagic flux until autolysosome formation. Moreover, VMP1 interacts with Cdt2, the adaptor subunit of the CRL4/Cdt2 E3 ligase complex, which is able to catalyze the VMP1 ubiquitination. ER: endoplasmic reticulum.

## Data Availability

All the generated data are compiled and given in MS or Appendix A. The raw data will be provided upon reasonable request.

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
