# Peer review of "Ubiquitination Is a Novel Post-Translational Modification of VMP1 in Autophagy of Human Tumor Cells"

_ijms, 2023, doi:10.3390/ijms241612981_

Round 1
Reviewer 1 Report
In the manuscript “Ubiquitination is a novel post-translational modification of VMP1 in autophagy of human tumor cells” Renna et al. study the ubiquitination of VMP1 and link it to autophagy and autophagosome turnover. VMP1 is an interesting transmembrane protein important for autophagy regulation. The group of M. Vaccaro performed many studies elucidating the role and function of VMP1 in autophagy. In the current manuscript, they show that VMP1 is ubiquitinated and that ubiquitinated VMP1 can be found at the nascent autophagosome and in autolysosomes arguing that ubiquitination of VMP1 is a constitutive process and that ubiquitinated VMP1 is present throughout the lifecycle of autophagosomes. They also identify a potential E3 ligase complex that might be involved in VMP1 ubiquitination. The physiological relevance of these events is not studied.
The manuscript is well written and nicely characterizes new molecular details of VMP1 biology. Before publication, few points should be addressed to underline the physiological occurrence of the observations.
Major points:
(1) In Figure 1 the authors only use exogenously expressed ubiquitin variants to study the ubiquitination of VMP1. It is known that overexpression of ubiquitin can induce the non-physiological modification of cellular proteins. Therefore, it is important that the authors also show that endogenous ubiquitin modifies VMP1. E.g. the authors could use their VMP1-V5 construct, perform and anti-V5 IP and then blot against endogenous ubiquitin (same antibody as used in Figure 3a).
(2) Figure 2: the authors conclude that VMP1 is not modified by Ub(K0). However, they IPed significantly less VMP1 in the respective experiment. Were the expression levels of Myc-Ub(WT) and Myc-Ub(K0) comparable? The Myc signal in inputs should be tested.
Minor points:
(1) Page 2, line 81: ubiquitination can also lead to autophagosomal protein degradation.
(2) Figure 4f, g: why are these panels not moved to Figure 5, following the argumentation in the text? The current panel numbering is confusing.
Text should be checked for grammar.
Reviewer 2 Report
The manuscript by Felipe J. Renna et al., reveals that ubiquitinated VMP1 regulates autophagy in human tumor cells and that is a newly identified post-translational modification of VMPI. The authors have performed series of experiments to obtain adequate data to confirm their hypothesis and showed interesting results. However, the review of this manuscript rises several questions about some findings and confusion about selecting cell lines for particular experiments.
The authors have mainly used two cell lines: non-cancerous cells HEK293T and cancerous cells HeLa in experiments. But what is the basis of selecting the cell lines for each experiment? One cell line was used in immunofluorescence labelling while other was used in immunoblotting or vise versa to prove the same hypothesis. For an example, figure 4 & 5 show involvement of ubiquitinated VMP1 in autophagic process. Hela cells were used in all the experiments in figure 4 & 5 except figure 5-a where HEK293T cells were used. Figure 7- a, e – Hela cells, Figure 7- c, h - HEK293T, Figure 1 – a -Hela cells, Figure 1 – b, c, d, e, f - HEK293T etc. This happens though out your manuscript. Therefore, please explain. I would like to suggest using one cell line in one set of data. As you have focused on tumor cells, it would be better to use cancer cell line.
Figure 1 – b: Starvation induced the autophagy. But there is no visible difference between basal and starved conditions. What did you expect to show with these results?
Figure 1- e IP: FIAG or FLAG?
Figure 3 – a shows the level of ubiquitin in VMP1-GFP transfected cells with or without treatment of MG132. It is clear that the level of ubiquitin was not affected by the proteosome inhibition. But in figure 3 – h the level of ubiquitin was not shown in the presence of CQ. Instead of that only the level of LC3 was shown. How do you explain the ubiquitination of VMP1 in presence of CQ with the expression of LC3? Why didn’t you show the level of ubiquitin with treatment of CQ? Even though, the authors explain the association between VMP1, Ub and LC3 later in the manuscript, readers can be confused with the sudden appearance of LC3.
The association between VMP1-Ub-LC3 was nicely shown by immunofluorescence labelling in figure 4 – a, b, c. But I would like to suggest it to be further confirmed by immunoblotting as it is a key finding in this research.
Round 2
Reviewer 1 Report
The authors adequately addressed all my concerns. Congratulations to a nice paper!